# A Cross-Sectional Study of Seroprevalence of Strongyloidiasis in Pregnant Women (Peruvian Amazon Basin)

**DOI:** 10.3390/pathogens9050348

**Published:** 2020-05-04

**Authors:** Sonia Ortiz-Martínez, José-Manuel Ramos-Rincón, María-Esteyner Vásquez-Chasnamote, Jhonatan J. Alarcón-Baldeón, Jorge Parraguez-de-la-Cruz, Olga-Nohelia Gamboa-Paredes, Patricia Schillyk-Guerra, Luis-Alfredo Espinoza-Venegas, Viviana-Vanessa Pinedo-Cancino, Ramón Perez-Tanoira, Miguel Górgolas-Hernández-Mora, Martin Casapía-Morales

**Affiliations:** 1Consultorio El Ballestero, Servicio de Salud Castellano Manchego, 2614 Albacete, Spain; soniaom1978@gmail.com; 2Departamento de Medicina Clínica, Universidad Miguel Hernández de Elche, 03550 Alicante, Spain; 3Servicio de Medicina Interna, Hospital General Universitario de Alicante, 03010 Alicante, Spain; 4Centro de Investigación de Recursos Naturales, Universidad Nacional de la Amazonia Peruana, 16001 Iquitos, Peru; vmariaesteyner.12@gmail.com; 5Laboratorio Clínico. Asociación Civil Selva Amazónica, 16001 Iquitos, Peru; jalarcon@acsaperu.org (J.J.A.-B.); jparraguez@acsaperu.org (J.P.-d.-l.-C.); pschillyk@acsaperu.org (P.S.-G.); 6Asistente de Investigación, Asociación Civil Selva Amazónica, 16001 Iquitos, Peru; ogamboa@acsaperu.org; 7Servicio de Enfermedades Infecciosas y Medicina Tropical, Hospital Regional de Loreto, 16001 Iquitos, Peru; laev16@hotmail.com; 8Laboratorio de Biología Molecular e Inmunología, Unidad Especializada del LIPNAA-CIRNA, Universidad Nacional de la Amazonia Peruana, 16001 Iquitos, Peru; vivicancino@gmail.com; 9Servicio de Microbiologia, Hospital Universitario Príncipe de Asturias, 28802 Alcalá de Henares, Spain; ramontanoira@hotmail.com; 10División de Enfermedades Infecciosas, Hospital Universitario Fundación Jiménez Díaz, 28040 Madrid, Spain; mgorgolas@me.com; 11Departamento de Medicina, Universidad Autónoma de Madrid, 28029 Madrid, Spain; 12Departamento Médico, Asociación Cívica Selva Amazónica, 16001 Iquitos, Peru; mcasapia@acsaperu.org; 13Facultad de Medicina, Universidad Nacional de la Amazonia Peruana, 496 Iquitos, Peru

**Keywords:** *Strongyloides stercoralis*, serology, seroprevalence, prevalence, Peru, Amazon

## Abstract

Strongyloidiasis is a soil-transmitted helminthiasis with a high global prevalence. Objectives: We aimed to evaluate the prevalence of *Strongyloides stercoralis* infection and assess strongyloidiasis serology as a screening technique in the Peruvian Amazon. Material and Methods: We performed a cross-sectional study of strongyloidiasis in 300 pregnant women in Iquitos (Peru) from 1 May 2019 to 15 June 2019. Women were tested using serology (Strongyloides IgG IVD-ELISA kit) as an index test and the modified Baermann technique and/or charcoal fecal culture as the parasitological reference standard. Results: The reference tests showed *S. stercoralis* in the stool of 30 women (prevalence: 10%; 95% confidence interval [CI] 7.1% to 13.9%), while 101 women tested positive on the blood test (prevalence: 33.7%; 95% CI 28.6% to 39.4%). Fourteen of the 15 women (93.3%) with positive results according to the modified Baermann technique, and 14 of the 23 women (56.5%) with positive charcoal cultures also had positive serological results. Serology showed a sensitivity of 63.3% and a negative predictive value of 94.4%. Conclusion: In Iquitos, pregnant women have a high prevalence of *S stercoralis*. *S. stercoralis* ELISA could be an excellent tool for population-based screening, as it has a high negative predictive value that can help to rule out the presence of active infection.

## 1. Introduction

Strongyloidiasis is an infection caused by the human parasitic roundworm *Strongyloides stercoralis* [1]. It is a soil-transmitted helminthiasis thought to affect some 370 million people worldwide [2,3]. The helminth is endemic to tropical and subtropical regions but can also occur in any area with an increased risk of fecal contamination due to poor sanitation or inadequate water supply, among other factors [1,2]. The number of people affected and the risk of infection varies among different population groups. For example, prevalence is higher in older people owing to autoinfection [4,5]. In pregnant women, particularly those with underlying conditions, strongyloidiasis can present in a more severe and disseminated form [6,7,8], which makes screening all the more relevant [9].

Several diagnostic methods are available for detecting strongyloidiasis, including direct stool smears, the Baermann technique, the Harada–Mori filter paper culture, charcoal cultures, and nutrient agar plate cultures. While all these methods have poor sensitivity—limiting their utility for describing the distribution, burden, and clinical characteristics of chronic infections—the first one is the least sensitive, and the last two are the most sensitive [10]. Molecular biological procedures such as polymerase chain reaction can be used as confirmatory tests [11]. Sensitivity improves with larger numbers of sequentially collected fecal samples [12].

Serological tests are a useful instrument for diagnosing strongyloidiasis in non-endemic areas, perhaps because of the presence of the chronic (asymptomatic) form of strongyloidiasis [13]. However, this method is cumbersome in endemic regions due to cross-reactivity with other helminthic diseases [13,14]. Using a recombinant antigen with an ELISA test or the luciferase immunoprecipitation system has resulted in a specificity of almost 100% for diagnosing *S. stercoralis* infections in non-endemic areas [13,14].

In Peru, the Ministry of Health reviewed the cross-sectional prevalence studies of *S. stercoralis* infection conducted in different areas of the country between 1981 and 2001 [15]. The mean prevalence was 6.6%, with variations according to location and diagnostic methods. Despite the high prevalence of the infection, there is limited knowledge of its epidemiology and sero-epidemiology [16]. To the best of our knowledge, there are no published sero-epidemiological studies on *S. stercoralis* in pregnant women. Indeed, there is a paucity of literature about *S. stercoralis* infection during pregnancy in general [6,7,9,17,18].

This study aimed to assess the prevalence of *S. stercoralis* infection in pregnant women and the value of serology as a population-based screening tool in the Peruvian Amazon Basin.

## 2. Methods

We performed a cross-sectional survey in an urban and periurban area in the Peruvian Amazon, using stool examinations and serologic testing for *S. stercoralis*, from 1 May 2019 to 15 June 2019.

### 2.1. Study Population

We included pregnant women attending the health centers in four districts of the greater Iquitos area, located in the Peruvian Amazon (Figure 1). Participants were selected through convenience sampling (i.e., on days when the researcher was at the health center) when they visited the midwife during prenatal check-ups.

### 2.2. Procedures

All patients provided a stool sample for identifying parasitic infections (*S. stercoralis* infection and other soil-transmitted helminths) and a blood sample for serology.

### 2.3. Stool Examination for S. stercoralis Infection

All fecal samples were processed using both the modified Baermann technique and a charcoal fecal culture. A stool was classified as positive for *S. stercoralis* if larvae were identified using either of these techniques. Stool examination was considered the reference standard diagnostic method.

Modified Baermann technique. Fecal specimens were processed using the usual method, described elsewhere [12]. Briefly, 5 g of fresh feces was placed at the center of a mesh sieve, which was partly immersed in a sedimentation flask containing water at 37 °C. The fecal specimens were left for 1 h at room temperature, inducing any larvae to migrate out of the fecal suspension into the warm water. The upper layer of the water was discarded by retaining 10 mL at the bottom of the funnels. The remaining fluid was transferred to a 15 mL test tube and centrifuged at 5000× *g* rpm for 5 min. The sediment was examined microscopically for the presence of larvae.

Charcoal culture. Fecal specimens were processed using the usual method [19]: 10 g of fresh fecal material was thoroughly mixed with distilled water and then with an equal quantity of granulated charcoal. The fecal–charcoal mixture was placed at the center of a Petri dish lined with moist filter paper. The Petri dish was sealed with vinyl tape and left in a dark room at 30 °C; on the seventh day, it was examined microscopically for *S. stercoralis* larvae.

### 2.4. Serology

Detection of *S. stercoralis* IgG antibodies was performed by enzyme-linked immunosorbent assay (ELISA) using the Strongyloides IgG IVD-ELISA kit (DRG Instruments GmbH, Marburg, Germany), approved by the European Community and featuring microtiter wells coated with a soluble fraction of *S. stercoralis* L3 filariform larval antigen. The procedure was performed in accordance with the manufacturer’s recommendations.

The technicians performing the stool examinations were different from those working in serology, and all technicians knew only the results of their own tests.

### 2.5. Stool Examination for Other Intestinal Helminthic Infestations

Fecal specimens were processed using the Kato–Katz technique [20]. Firstly, a template with a hole was placed on a slide. Secondly, the fecal sample was sieved and then spread over the hole. Lastly, the template was lifted off, and the sample remaining on the slide was covered with a strip of glycerol-soaked cellophane (to clear the fecal material from around the eggs). The sample was screened for helminths other than *S. stercoralis* to obtain data on coinfection of *S. stercoralis* with other parasitic worms and to determine how this coinfection could affect the specificity of serology.

### 2.6. Data Analysis

The collected data were systematically recorded and analyzed using IBM SPSS Statistics version 22.0. We used the chi-square test or Fisher’s exact test to determine the presence of *S. stercoralis* by several demographic variables, considering results to be significant when the *p* value was less than 0.05.

We used the Cohen’s kappa test to establish the degree of agreement between parasitological methods (charcoal culture and modified Baermann’s technique). To obtain the 95% confidence interval (CI) for *S. stercoralis* infection, we applied the method described by Newcombe [21]. Sensitivity, specificity, positive predictive value, and negative predictive value, plus their respective 95% CIs, were calculated for ELISA serology of *S. stercoralis* infection using the modified Baermann technique and/or charcoal culture as the reference standard.

### 2.7. Ethical Considerations

The Ethics Committee of the General University Hospital of Alicante (Spain) approved the project (PI2018/113), as did the Ethics Committee of Loreto Regional Hospital in Iquitos (Peru) (027-CIEI-HRL-2019). After receiving information on the study, individuals who volunteered to participate gave their written consent and were included. All results were kept strictly confidential, and individuals who were positive for intestinal parasites were treated free of charge.

## 3. Results

### 3.1. Study Population

The study included 300 pregnant women with a mean age of 26.7 years (standard deviation 6.4; range 13 to 38). The epidemiological data are shown in Table 1.

### 3.2. Stool Examination of S. stercoralis Infection

The modified Baermann technique and/or charcoal culture yielded positive results in 30 participants (prevalence 10%; 95% CI 7.1% to 13.9%). Seven cases were diagnosed based on results from the modified Baermann technique alone, 15 cases based on the charcoal culture alone, and 8 cases based on positive results of both tests. The kappa index, measuring the concordance between the two methods, was 0.384.

### 3.3. Intestinal Helminthic Infestations in Stool

The Kato–Katz technique showed positive results in 68 women (22.7%): 49 were positive for one parasite (16.3%), 18 for two (6.0%), and 1 for three (0.3%). The parasitic pathogens identified were *Ascaris lumbricoides* (n = 49; 16.3%), *Trichuris trichiura* (n = 27; 9.0%), hookworm (n = 11; 3.7%), and *Hymenolepis nana* (n = 1; 0.3%).

### 3.4. Co-Infections of S. stercoralis and Other Helminths 

Of the 30 participants who were positive for *S. stercoralis* infection, 14 were infected with other helminthic parasites. Table 2 shows *S. stercoralis* co-infections.

### 3.5. Serology of S. stercoralis Infection

Based on a commercial density index value of 1.1, ELISA was positive in 101 cases (prevalence: 33.7%; 95% CI 28.6% to 39.4%). Most participants (n/N = 14/15; 93.3%) testing positive according to the modified Baermann technique were also positive for *S. stercoralis* on serology (*p* < 0.001). Similarly, 56.5% of participants with a positive charcoal fecal culture had a positive blood test (*p* = 0.016). Serological results were also positive in 63.3% of the participants with two positive reference tests (*p* < 0.001). The presence of hookworm, *A. lumbricoides*, and *T. trichiura* infections was not significantly associated with a *S. stercoralis-*positive ELISA (Table 3).

### 3.6. Participant and Housing Characteristics and Their Relation to S. stercoralis Infection

Women living in rural areas and in houses with leaf roofing were more likely to be infected with *S. stercoralis*, according to the ELISA results (*p* = 0.007 and *p* = 0.033, respectively). The detection of *S. stercoralis* was not correlated with age, number of bowel movements, or anemia (Table 4).

### 3.7. Serology Versus Stool Examination for Detecting S. stercoralis

We compared the stool and ELISA results for the 300 participants. Using the modified Baermann technique and/or charcoal culture as the parasitological reference standard, the ELISA test showed a sensitivity of 63.3% and a negative predictive value of 94.4% (Table 5).

## 4. Discussion

In our study, infection with *S. stercoralis* was observed in 1 of every 10 pregnant women living in the greater Iquitos area (Peru), according to results from the modified Baermann technique and charcoal agar culture, and in 1 of every 3 women according to the serology results. Literature on *S. stercoralis* infection during pregnancy is scarce [6,7,8,9,17,18], and there are few studies examining the impact of *S. stercoralis* infection on mother and fetus [22,23]. One study suggests that it has no direct effect on maternal or fetal outcomes, since the helminth does not cross the placenta [22]. There are several case reports of severe *Strongyloides* infection during pregnancy in immunocompromised women (with human T-cell leukemia virus type [HTLV] co-infection, on steroids, etc.) [6,8]. Our study provides data on the prevalence of *S. stercoralis* in pregnant women.

The prevalence estimate observed in pregnant women is similar to that reported by Yori et al., corresponding to 8.7%, [16] in a similar study using stool examination near Iquitos (Nanai river) and to the prevalence observed by Errea et al. [24] in children in Padre Cocha, also near Iquitos. However, our estimate is lower than that of older studies performed in the higher Amazon river, where prevalence was estimated at 16% in schoolchildren in San Martin in 1999 [25], at 19.5% in outpatients with diarrhea in Madre de Dios in 2001 [26], and at 38.5% in a rural community of the Pasco region in 2005 [27]

The ELISA used in this study has been available for a number of years, but cross-reactivity with filarial species, which are often co-endemic with *S. stercoralis* [16,28,29], has restricted the use of this technique. There is evidence that an ELISA based on a recombinant antigen has a high sensitivity and no cross-reactivity with *Onchocerca volvulus* or *Loa* [29]; however, there is still a risk of cross-reactivity with hookworms and other helminth infections [30].

Serology, alone or in combination with stool testing, has been used for screening, diagnosis, and post-treatment monitoring in immigrants from endemic areas living in non-endemic countries [13,30]. This method is considered appropriate for screening due to its higher sensitivity compared to stool examination [13,31]. In our study, the sensitivity of *Strongyloides* serology was low. This could be because *Strongyloides* serology measures IgG antibody response to the presence of *S. stercoralis,* and although it is useful for diagnosing chronic strongyloidiasis in non-endemic areas [13,14], it cannot detect newly acquired infections in which the IgG immune response has not yet increased.

Different studies have analyzed the utility of serology for diagnosing *S. stercoralis* in endemic countries. In South America, several sero-epidemiological studies have taken place, mainly in Argentina, Peru, and Ecuador. In Argentina, a 2010 study by Krolewiecki et al. [32] assessed the sensitivity and specificity of four *Strongyloides* serological tests (ELISA based on crude antigen, ELISA with recombinant antigen, luciferase immunoprecipitation system assay based on a 31 kDa recombinant antigen, and recombinant *S. stercoralis* immune-reactive antigen), detecting a prevalence of 29.4% in 228 stool samples. In all cases, sensitivity exceeded 84%, and specificity was 100%.

In our study, the ELISA test was positive in 33.7% of the women tested and showed a high negative predictive value. These results are very similar to those reported by Yori et al. [16] in their survey in a rural community of the Peruvian Amazon. The prevalence of *S. stercoralis* infection in that study was of 8.7%, but serology was positive in 72%, with a high negative predictive value.

Echazú et al. [33] performed a community-based pragmatic study in Northwestern Argentina. In 2012, they observed a prevalence of 51%; after three years of treatment of the community with albendazole and ivermectin (in 2015), this figure dropped to 14%. In Ecuador, Anselmi et al. [28] used an epidemiological approach to parasitic infection screening in a remote community (n = 705), finding *S. stercoralis* larvae in 4.3% of the samples, whereas serology was positive in 22.7% of the individuals tested. This study was similar to ours, with a difference of prevalence in stool detection and seroprevalence of about 20%.

Outside Latin America, in Southeast Asia and the Pacific, Akiyama and Brown reported a seroprevalence of 16% in 475 residents of Hawaii; the largest proportion of seropositive cases was found in Micronesians (30%) [34]. In Vietnam, Diep et al. [35] observed a sero-reactivity to *S. stercoralis* in 29.1% of adults (N = 1340) and 5.5% of children (N = 270); men were more frequently sero-reactive when living in rural central highlands. In Kuala Lumpur (Malaysia), Zueter et al. [36] found a seroprevalence for *S. stercoralis* of 4.2% in cancer patients (N = 195) and 2.7% in healthy controls (N = 145). However, in another study in 54 adults being treated with corticosteroids, the seroprevalence was 31.5% [37]. A study conducted in 177 people in a leper colony in Thailand reported that 45% of the patients were seropositive, while 1% presented *S. stercoralis* in stool [37]. Sultan et al. [38] studied the prevalence of *S. stercoralis* in 147 residents of an underserved urban community of Dhaka (Bangladesh), observing a prevalence of 23.4% by Harada–Mori culture and of 61.2% by serology IgG4 against *S. stercoralis*.

In Australia, Hays et al. [39] undertook a three-year cohort study in 259 adult Aboriginals living in a remote northern community. The initial prevalence of 35.3% decreased to 5.8% after three years of community treatment with ivermectin. This study and that of Echazú et al. [33], both conducted in areas endemic for *S. stercoralis* infection, support the utility of serology for treatment follow-up in endemic rural areas.

In the Middle East, the *S. stercoralis* antibody test was positive in 4% of 100 immunosuppressed patients studied in Turkey [40].

All the studies here reviewed show the relevance of serology for the screening of *S. stercoralis* infection.

In Africa, we did not identify studies of sero-prevalence indexed in PubMed. While we found several studies using serological methods for the analysis of immigrants or adopted children living in Europe, none took place in an African country [41].

This study has several limitations. Firstly, we did not collect information about risk factors related to *S. stercoralis* infection, such as walking barefoot, bathing in rivers/streams, using municipal sewage, and other socioeconomic sanitary factors. Secondly, we studied a single stool specimen. The use of three stool samples would have given a higher positivity rate; however, two different techniques were used for the stool detection of larvae. Finally, the use of other techniques such as sedimentation concentration or Harada–Mori with more than one stool sample would have led to a higher diagnostic yield (increasing the overall sensitivity).

## 5. Conclusions

In conclusion, the prevalence of *S. stercoralis* infection in pregnant women in the city of Iquitos was about 10% when combining the modified Baermann method and charcoal culture, while it reached 30% when ELISA was used to analyze serum samples. Given the high negative predictive value of serology, the ELISA test is a highly appropriate screening tool. Additional sero-epidemiological studies are necessary to validate this test in rural regions, and more studies in pregnant women are crucial, given the increased risk of *S. stercoralis* infection in this population.

## Figures and Tables

**Figure 1 pathogens-09-00348-f001:**
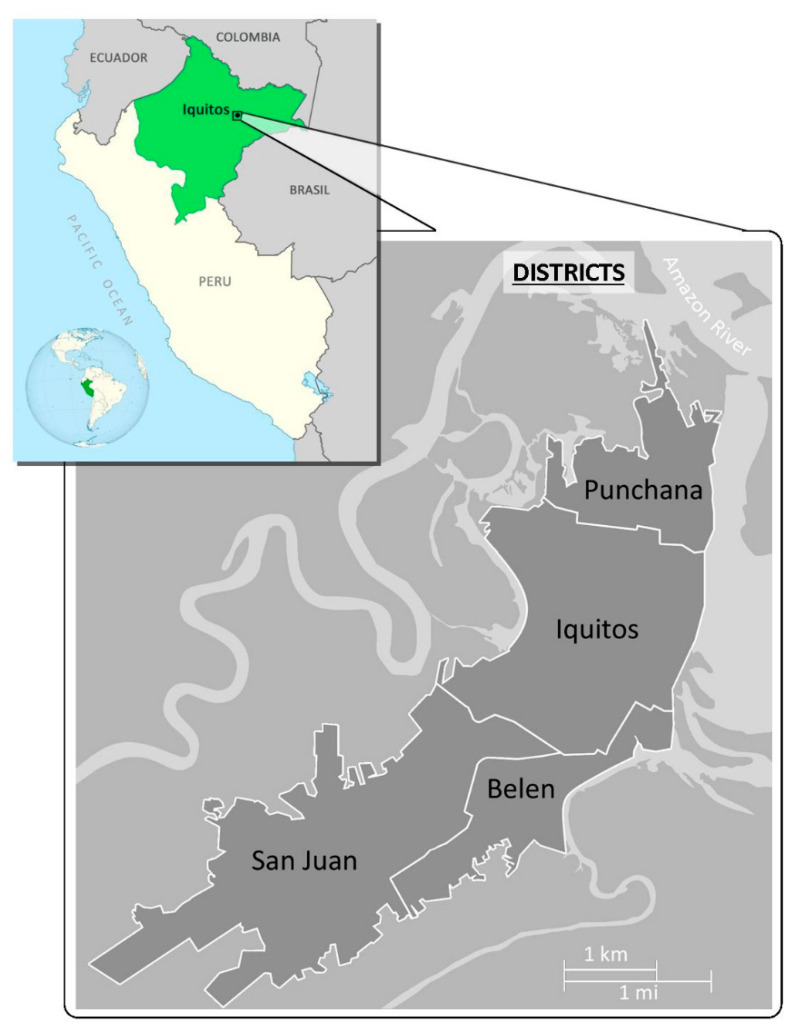
Map of Peru, with greater Iquitos highlighted.

**Table 1 pathogens-09-00348-t001:** Epidemiological characteristics of the participants (N = 300).

Variables	n (%)
Age, mean (SD)	26.7 (6.4)
Number of deliveries, mean (SD)	2.9 (1.7)
Gestation age in days, mean (SD)	172 (59)
Health center districts, n (%)	
San Juan	149 (49.7)
Puchana	79 (26.3)
Belén	51 (17.0)
Iquitos	21 (7.0)
Area of residence, n (%)	
Urban area	166 (55.3)
Rural area	134 (44.7)
Bowel movements/day, mean (SD)	1.5 (0.7)

SD: standard deviation.

**Table 2 pathogens-09-00348-t002:** *Strongyloides stercoralis* and co-infections (N = 30).

Pathogens	n (%)
*Strongyloides stercoralis* alone	16 (53.3)
*S. stercoralis +* hookworm	5 (16.7)
*S. stercoralis + Trichuris trichiura*	5 (16.7)
*S. stercoralis + Ascaris lumbricoides + T. trichiura*	1 (3.3)
*S. stercoralis + A. lumbricoides* + hookworms	1 (3.3)
*S. stercoralis + T. trichiura* + hookworms	1 (3.3)
*S. stercoralis + A. lumbricoides + T. trichiura* + hookworms	1 (3.3)

**Table 3 pathogens-09-00348-t003:** Results of *S. stercoralis* serology from stool examination using different techniques (modified Baermann method, charcoal culture, Kato–Katz technique).

Helminthic Parasites	*S. stercoralis* ELISA		*p* Value *
	Positive n (% row)	Negative n (% row)	
Stool examination for *S. stercoralis*			
Modified Baermann method positive (N = 15)	14 (93.3)	1 (6.7)	<0.001
Charcoal culture-positive (N = 23)	23 (56.5)	10 (43.5)	0.016
Modified Baermann method and/or charcoal culture-positive (N = 30)	19 (63.3)	11 (36.7)	< 0.001
Stool examination for other intestinal helminthic infestations			
*A. lumbricoides* (N = 49)	17 (34.7)	32 (65.3)	0.87
*T. trichiura* (N = 27)	27 (44.4)	15 (55.6)	0.21
Hookworms (N = 11)	11 (54.5)	5 (45.5)	0.13
*Hymenolepis nana* (N = 1)	1(100)	0 (0.0)	0.32

* The statistical method used was Chi square test/Fisher’s exact test.

**Table 4 pathogens-09-00348-t004:** Participant and housing characteristics of pregnant women with *S. stercoralis* detected in stool and by ELISA (N = 30).

Variables	*S. stercoralis* in Feces			*S. stercoralis* ELISA		
	Positive	Negative	*p* Value *	Positive	Negative	*p* Value *
Age, mean (SD)	26.7 (6.5)	26.9 (5.7)	0.84	26.7 (6.3)	26.7 (6.7)	0.35
Rural residence, n (%)	16 (53.3)	118 (43.7)	0.31	56 (55.4)	78 (39.2)	0.007
Characteristics of house, n (%)						
Dirt floor	8 (26.7)	81 (30.0)	0.70	30 (29.7)	59 (29.6)	0.99
Leaf roofing	27 (90)	207 (76.7)	0.094	86 (85.1)	148 (74.4)	0.033
Wooden house	29 (96.7)	257 (95.2)	0.71	98 (97.0)	188 (94.5)	0.32
Brick house	18 (60.0)	171 (63.3)	0.72	67 (66.3)	122 (61.3)	0.30
Characteristics of individuals						
Bowel movements/day, mean (SD)	1.5 (0.8)	1.5 (0.8)	0.95	1.6 (7.8)	1.5 (0.7)	0.64
Hemoglobin < 11 g/dL, n (%)	12 (40)	121 (44.8)	0.70	42 (41.6)	91 (45.7)	0.49

* The statistical method used was Chi square test/Fisher’s exact test. SD: standard deviation.

**Table 5 pathogens-09-00348-t005:** Comparison of *S. stercoralis* serological and stool examinations.

*S. stercoralis* ELISA	Modified Baermann Methods and/or Charcoal Culture		Total
	Positive	Negative	
Positive	19	82	101
Negative	11	188	199
Total	30	270	300
Sensitivity: 63.3% (95% CI 43.9% to 79.4%) Specificity: 69.6% (95% CI 63.7% to 74.9%) Positive predictive value: 18.8% (95% CI 11.9% to 28.1%) Negative predictive value: 94.4% (95% CI 90.0% to 97.1%)			

CI: confidence interval.

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
