# Peer review of "A Cross-Sectional Study of Seroprevalence of Strongyloidiasis in Pregnant Women (Peruvian Amazon Basin)"

_pathogens, 2020, doi:10.3390/pathogens9050348_

Round 1
Reviewer 1 Report
This study is an important contribution to information on prevalence of S. stercoralis and use of serology in screening for strongyloidiasis in an endemic region.
Strongyloides serology is a measure of the IgG antibody response to the presence of S. stercoralis, and is most useful for diagnosing chronic strongyloidiasis. Could the sensitivity be lower due to newly acquired infections in an endemic area and the IgG immune response not yet increased?
The following comments are recommendations to improve the final publication.
Line 31- Objective.
Line 36: replace "gold" with "reference" as a gold standard should represent true disease but the sensitivity is too low to represent true disease.
Line 66: please remove the old assumption 'positive reactions from previous infections and" as serology is also a measure of effectiveness of treatment and a post-treatment negative serology is a marker of previous infection.
Line 74: suggest replace "in special" with 'especially'
Line 94: Recommend remove 'gold' and replace with 'reference'

Author Response
Response to Reviewer 1
This study is an important contribution to information on prevalence of S. stercoralis and use of serology in screening for strongyloidiasis in an endemic region. Strongyloides serology is a measure of the IgG antibody response to the presence of S. stercoralis, and is most useful for diagnosing chronic strongyloidiasis. Could the sensitivity be lower due to newly acquired infections in an endemic area and the IgG immune response not yet increased?
Response: Thank you to the reviewer for the suggestion.
In this study the sensibility of the procedure was less than other studies, because you say a to newly acquired infections in an endemic area and the IgG immune response not yet increased.
And other cause of less sensibility is that we analyses only one feces sample and using two or three samples it is increase de sensibility.
Response: We have included your hypothesis for low sensibility in the discussion section:
… In our study, the sensitivity of Strongyloides serology was low. This could be because Strongyloides serology measures IgG antibody response to the presence of S. stercoralis, and although it is useful for diagnosing chronic strongyloidiasis in non-endemic areas [13,14], it cannot detect newly acquired infections in which the IgG immune response has not yet increased.
The following comments are recommendations to improve the final publication.
Line 31- Objective. suggest replace 'gold' with 'reference' as parasitological confirmation confirms disease, but a negative stool does not mean the person does not have 'true disease'.
Response: Thank you to the reviewer for raising this relevant point. We have changed “gold standard” to “parasitological reference standard”.
Line 36: replace "gold" with "reference" as a gold standard should represent true disease but the sensitivity is too low to represent true disease.
Response: Thank you for pointing this out.
Line 66: please remove the old assumption 'positive reactions from previous infections and" as serology is also a measure of effectiveness of treatment and a post-treatment negative serology is a marker of previous infection.
Response: Thank you for noticing this error. We have removed ”positive reactions from previous infections and”
Line 74: suggest replace "in special" with 'especially'
Response: This paragraph has been revised.
Line 94: Recommend remove 'gold' and replace with 'reference
Response: Thank you, we have changed this term.
Line 106. . . Are you able to provide the temperature as S. stercoralis are reputed to only survive in certain temperatures
Response: Thank you, we have included the temperature ( 30ºC).
Line 113…
Response: We have changed it.
Line 119…
Response: Changed.
Table 5
Response: We have rewritten Strongyloides stercoralis in italics
Table 6.
Response: We have reviewed the consistent manning of the test in this table, if it is the same test.

Reviewer 2 Report
This paper titled “A cross-sectional study of seroprevalence of strongyloidiasis in pregnant women (Peruvian 4 Amazon Basin) and systematic literature review of strongyloidiasis in Peru” used two types of diagnostic tests, microscopy and serology, to diagnose and determine the prevalence of strongyloidiasis in pregnant women. For microscopical examination, the authors used a combination of Baermann technique and charcoal culture, and for serology they used ELISA test. The authors also undertook a literature review of strongyloidiasis cases happened in Peru from 1981 to 2019. It is understood that authors had three aims:
- To identify the prevalence of strongyloidiasis in pregnant women
- To evaluate the sensitivity of serology test, ELISA, in diagnosing strongyloidiasis.
- To assess the epidemiology of stercoralis infection in Peru by performing systematic literature review of strongyloidiasis cases from 1981 to 2019.
It is an interesting study that would contribute to the current knowledge about Strongyloides prevalence in different groups, e.g. pregnant women. However, there are several shortcomings and potential areas for improvement, and changes need to be done before this paper can be considered for publication.
The authors should consider the following points that will help them to improve their manuscript:
- It would be recommended to conduct and publish Aim 3 of this study – assessment of the epidemiology of stercoralis infection in Peru – as a separate study. First of all, the literature review conducted for this study does not tell us about the epidemiology of strongyloidiasis that authors aimed for. Second of all, including the literature review in this study makes it hard to follow the structure of the manuscript and its aims.
- Authors should describe more carefully which samples they analysed using Baermann and/or culture technique. How did serology perform compared to these microscopic tests?
- Authors should take care on explaining whether identifying the prevalence of other helminths and cross-infection was also an aim of this study? Or was it done to assess the specificity of diagnostic methods? Authors need to make it clearer stating in the introduction. How did the co-infection impact on the sensitivity of serology and/or microscopic tests for strongyloidiasis.
- The result section needs to be widely revised and results presented in a clearer way. The data presented in some tables is very hard to follow, e.g. Table 3.
Further, the authors should consider and address the following points that will help them to improve the quality of the study and manuscript.
Introduction:
Lines 56-57: I would cite more studies supporting the statement that Strongyloides prevalence increases in older age group. There has been only one study in Vietnam that showed the increase of infection prevalence with age. However Vietnamese veterans are known to be at a high risk of strongyloidiasis and also study didn’t look at other infections that could also be risk factors for Strongyloides infection. Generally, kids are more likely to get infected with soil-transmitted helminths.
Lines 57-58: Here authors state that strongyloidiasis can take a severe form in pregnant women citing only one case (Ref 6). This study (ref 6) found strongyloidiasis in a pregnant woman who was also confirmed to be infected with HLTV-1. This is important because HLTV-1 is a known risk factorfor severe strongyloidiasis. I would suggest citing more studies that show severe strongyloidiasis in pregnant women, or rephrase the sentence.
Line 60: Need to be more specific about “diagnostic methods that have poor sensitivity”. Which one or all of them?
Lines 60-64: This paragraph is a little confusing. Is it about all available diagnostic tests or only stool tests? Authors need to accurately describe which methods are more or less sensitive and/or specific for detection of different forms of Strongyloides infection (chronic, hyperinfection, disseminated).
Lines 65-66: Can authors please clarify why it is that serology is useful in diagnosing strongyloidiasis in non-endemic areas? (perhaps because of the chronic form of strongyloidiasis?)
Lines 70-71: Cite ref 10 after the first sentence.
Methods:
Line 91: add “other” before “soil-transmitted helminths”
Lines 93-95: Can authors please specify which faecal samples were processed using Baermann technique and which using charcoal culture. And what was the rational for choosing the test?
Lines 106-107: Did authors mean “at the seventh day”?
Results:
Lines 158-162: From reading this paragraph I understand that all faecal samples (300) were processed using both Baermann and charcoal culture. If this is true, then it contradicts with what is written in the methods section.
Line 180: Can authors please think of a different title to the Table 3.
Line 180: Table 3 is not very clear.
Discussion:
Lines 227-232: Because the aim of this study was to assess the prevalence of strongyloidiasis in pregnant women it would be more accurate to compare the results of this study with other studies on pregnant women.
Lines 273-277: Authors need to write what exactly two studies looked at. Also, it is not clear which areas are endemic and which are not endemic and for which disease?

Author Response
Response to Reviewer 2
This paper titled “A cross-sectional study of seroprevalence of strongyloidiasis in pregnant women (Peruvian 4 Amazon Basin) and systematic literature review of strongyloidiasis in Peru” used two types of diagnostic tests, microscopy and serology, to diagnose and determine the prevalence of strongyloidiasis in pregnant women. For microscopical examination, the authors used a combination of Baermann technique and charcoal culture, and for serology they used ELISA test. The authors also undertook a literature review of strongyloidiasis cases happened in Peru from 1981 to 2019. It is understood that authors had three aims:
- To identify the prevalence of strongyloidiasis in pregnant women
- To evaluate the sensitivity of serology test, ELISA, in diagnosing strongyloidiasis.
- To assess the epidemiology of stercoralis infection in Peru by performing systematic literature review of strongyloidiasis cases from 1981 to 2019.
It is an interesting study that would contribute to the current knowledge about Strongyloides prevalence in different groups, e.g. pregnant women. However, there are several shortcomings and potential areas for improvement, and changes need to be done before this paper can be considered for publication.
The authors should consider the following points that will help them to improve their manuscript:
1.It would be recommended to conduct and publish Aim 3 of this study – assessment of the epidemiology of stercoralis infection in Peru – as a separate study. First of all, the literature review conducted for this study does not tell us about the epidemiology of strongyloidiasis that authors aimed for. Second of all, including the literature review in this study makes it hard to follow the structure of the manuscript and its aims.
Response: Thank you for your comments. Including the literature review may make the manuscript harder to follow, but it gives an overall picture of strongyloidiasis in Peru, which we consider to be relevant to our study.
2.Authors should describe more carefully which samples they analysed using Baermann and/or culture technique. How did serology perform compared to these microscopic tests?
Response: Thank you for pointing this out. We have included the following sentence in section 2.3:
“All fecal samples were processed using both the modified Baermann technique and a charcoal fecal culture”.
…and the following sentence in section 2.4:
“The technicians performing the stool examinations were different from those working in serology, and all technicians knew only the results of their own tests.”
3.Authors should take care on explaining whether identifying the prevalence of other helminths and cross-infection was also an aim of this study? Or was it done to assess the specificity of diagnostic methods? Authors need to make it clearer stating in the introduction. How did the co-infection impact on the sensitivity of serology and/or microscopic tests for strongyloidiasis.
Response: in accordance with the reviewer’s suggestion, we have added the following sentence in section 2.5:
“The sample was screened for helminths other than S. stercoralis to obtain data on coinfection of S. stercoralis with other parasitic worms, and to determine how this coinfection could affect the specificity of serology.”
3.The result section needs to be widely revised and results presented in a clearer way. The data presented in some tables is very hard to follow, e.g. Table 3.
Response: We have revised the results
Further, the authors should consider and address the following points that will help them to improve the quality of the study and manuscript.
Introduction:
Lines 56-57: I would cite more studies supporting the statement that Strongyloides prevalence increases in older age group. There has been only one study in Vietnam that showed the increase of infection prevalence with age. However Vietnamese veterans are known to be at a high risk of strongyloidiasis and also study didn’t look at other infections that could also be risk factors for Strongyloides infection. Generally, kids are more likely to get infected with soil-transmitted helminths.
Response: Children are more likely to get infected with soil-transmitted helminths, but S. stercoralis infections increase with age owing to autoinfection (or endogenous reinfection) and because adults working in agriculture are in close contact with the parasites living in the soil, which can penetrate the skin.
This paragraph has been revised, and includes the sentence “For example, prevalence is higher in older people owing to autoinfection”.
Lines 57-58: Here authors state that strongyloidiasis can take a severe form in pregnant women citing only one case (Ref 6). This study (ref 6) found strongyloidiasis in a pregnant woman who was also confirmed to be infected with HLTV-1. This is important because HLTV-1 is a known risk factor for severe strongyloidiasis. I would suggest citing more studies that show severe strongyloidiasis in pregnant women, or rephrase the sentence.
Response: In view of this comment, we have cited more studies that show severe strongyloidiasis in pregnant women (Buresch AM, Judge NE, Dayal AK, Garry DJ. A Fatal Case of Strongyloidiasis in Pregnancy. Obstet Gynecol. 2015;126(1):87–89. doi:10.1097/AOG.0000000000000676;
Hays R, McDermott R. Strongyloides stercoralis infection and antenatal care. Med J Aust. 2015;203(1):18–19. doi:10.5694/mja15.00429)
Furthermore:
- We have modified the sentence as follows: “In pregnant women, particularly those with underlying conditions, strongyloidiasis can present in a more severe and disseminated form [6,7,8], which makes screening all the more relevant [9].”
- We have added information in paragraph 4 of section 1.
- We have rewritten the first paragraph of the discussion to include information about S. stercoralis infection in pregnancy.
- And we have included a consideration in the conclusion section about S. stercoralis infection in pregnancy, noting that more studies are needed.
Line 60: Need to be more specific about “diagnostic methods that have poor sensitivity”. Which one or all of them?
Lines 60-64: This paragraph is a little confusing. Is it about all available diagnostic tests or only stool tests? Authors need to accurately describe which methods are more or less sensitive and/or specific for detection of different forms of Strongyloides infection (chronic, hyperinfection, disseminated).
Response: in accordance with the reviewer’s suggestion, we have rewritten this paragraph as follows:
Several diagnostic methods are available for detecting strongyloidiasis, including direct stool smears, the Baermann technique, the Harada-Mori filter paper culture, charcoal cultures, and nutrient agar plate cultures. While all these methods have poor sensitivity—limiting their utility for describing the distribution, burden and clinical characteristics of chronic infections—the first is the least sensitive and the last two are the most sensitive [10]. Molecular biological procedures such as polymerase chain reaction can be used as confirmatory tests [11]. Sensitivity improves with larger numbers of sequentially collected fecal samples [12].
Lines 65-66: Can authors please clarify why it is that serology is useful in diagnosing strongyloidiasis in non-endemic areas? (perhaps because of the chronic form of strongyloidiasis?)
Response: Thank you for this comment. We have changed the sentence, which now reads:
“Serological tests are a useful instrument for diagnosing strongyloidiasis in non-endemic areas, perhaps because of the chronic (asymptomatic) form of strongyloidiasis [13].”
Lines 70-71: Cite ref 10 after the first sentence.
Response: Thank you for this observation, we have added the reference.
Methods:
Line 91: add “other” before “soil-transmitted helminths”
Response: Thank you, we have added “other”.
Lines 93-95: Can authors please specify which faecal samples were processed using Baermann technique and which using charcoal culture. And what was the rational for choosing the test?
Response: Thank you for this comment. We have added the following sentence:
“All fecal samples were processed using both the modified Baermann technique and a charcoal fecal culture”
Lines 106-107: Did authors mean “at the seventh day”?
Response: Thank you, we have changed this to “on the seventh day”.
Results:
Lines 158-162: From reading this paragraph I understand that all faecal samples (300) were processed using both Baermann and charcoal culture. If this is true, then it contradicts with what is written in the methods section.
Response: We have included in methods section “All fecal samples were processed using both the modified Baermann technique and a charcoal fecal culture.”
Line 180: Can authors please think of a different title to the Table 3.
Response: We have changed the title of table 3.
Line 180: Table 3 is not very clear.
Response: We have modified table 3 to make it clearer.
Discussion:
Lines 227-232: Because the aim of this study was to assess the prevalence of strongyloidiasis in pregnant women it would be more accurate to compare the results of this study with other studies on pregnant women.
Response: Thank you for the recommendation. This paragraph now reads:
Infection with S. stercoralis was observed in 1 of every 10 pregnant women living in the greater Iquitos area (Peru) according to results from the modified Baermann technique and charcoal agar culture, and in 1 of every 3 women according to the serology results. Literature on S. stercoralis infection during pregnancy is scarce [6,7,8,9,17,18], and there are few studies examining the impact of S. stercoralis infection on mother and fetus [40,41]. One study suggests that it has no direct effect on maternal or fetal outcomes, since the helminth does not cross the placenta [40]. There are several case reports of severe Strongyloides infection during pregnancy in immunocompromised women (those with HTLV co-infection, those on steroids, etc.) [6,8]. Our study provides data on the prevalence of S. stercoralis in pregnant women.
Lines 273-277: Authors need to write what exactly two studies looked at. Also, it is not clear which areas are endemic and which are not endemic and for which disease?
Response: thank you, we have modified this paragraph in accordance with your comment.

Reviewer 3 Report
Overall a well conducted study and literature review. There are three points that I believe must be addressed to ensure correct interpretation of the results.
line 76 "screening tool": here and elsewhere in the manuscript there seems to be some lack of clarity if the intention is to use serology as a population based screening tool or as a screening tool for individuals where it could be used as the first step of a two step diagnostic protocol. Please clarify and ensure consistent use throughout manuscript.
Section 3.7: the result of Baermann’s alone is presented with the same weight as the combined Baermann’s and/or charcoal. I believe this is misleading. The methods clearly state that the combined result is the primary outcome and this needs to be presented with greater prominence. The performance of serology as a screening test for individual diagnosis is not optimal when compared to the combined method and focusing on the results of Baermann alone gives a false impression.
lines 161-162 It is not clear what the intended interpretation of the Cohen's kappa index result is. The value of 0.384 is generally considered to indicate minimal agreement; the p value is not particularly informative.
In addition, the manuscript may benefit from some minor amendments to aid clarity:
lines 73-74 "in special" could be "especially" or "in particular"
line 109 should read ... enzyme linked immunosorbent assay...
line 113 a mean value is mentioned: is this the mean of repeated readings on a single well or were samples run in duplicate/triplicate?
line 134 "properly" is meaningless in this context. Would "systematically" convey the intended meaning?
table 1 "depositions per day" may be better as "bowel movements per day" or "defecations per day"
line 183 p values given here both appear to be for ELISA (as per table 4) and not to ELISA and stool respectively
Table 3: please indicate in a footnote which test has been used to generate p values – I presume these are Chi square/Fisher’s exact but the statistical methods outline that Cohen’s kappa is used to compare tests
Author Response
Response to Reviewer 3
Comments and Suggestions for Authors
Overall a well conducted study and literature review. There are three points that I believe must be addressed to ensure correct interpretation of the results.
1.line 76 "screening tool": here and elsewhere in the manuscript there seems to be some lack of clarity if the intention is to use serology as a population based screening tool or as a screening tool for individuals where it could be used as the first step of a two step diagnostic protocol. Please clarify and ensure consistent use throughout manuscript.
Response: Thank you for this comment. In our study serology was used as a population-based screening tool. We have included this information in line 76.
2.Section 3.7: the result of Baermann’s alone is presented with the same weight as the combined Baermann’s and/or charcoal. I believe this is misleading. The methods clearly state that the combined result is the primary outcome and this needs to be presented with greater prominence. The performance of serology as a screening test for individual diagnosis is not optimal when compared to the combined method and focusing on the results of Baermann alone gives a false impression.
Response: In accordance with the reviewer’s comment, we have changed all Baerman method references to modified Baerman method
We have removed the sentence ”When we compared the ELISA test to only the modified Baermann method, the sensitivity for the serological test was 93.3%, and the negative predictive value was 99.4% (Table 5).”
And we have removed a reference to “Only modified Baerman method” in table 5.
lines 161-162 It is not clear what the intended interpretation of the Cohen's kappa index result is. The value of 0.384 is generally considered to indicate minimal agreement; the p value is not particularly informative.
Response: We agree with the reviewer that the P value is not particularly informative and we have removed it from this sentence.
3.In addition, the manuscript may benefit from some minor amendments to aid clarity:
lines 73-74 "in special" could be "especially" or "in particular"
Response: Thank you, we have changed it.
line 109 should read ... enzyme linked immunosorbent assay...
Response: Thank you, we have changed it.
line 113 a mean value is mentioned: is this the mean of repeated readings on a single well or were samples run in duplicate/triplicate?
Response: Thank you for this question. It was performed on a single well and samples were run once or in some situations twice. We have revised this whole paragraph, which now reads:
“Detection of S. stercoralis IgG antibodies was performed by enzyme-linked immunosorbent assay (ELISA) using the Strongyloides IgG IVD-ELISA kit (DRG Instruments GmbH, Marburg, Germany), approved by the European Community and featuring microtiter wells coated with a soluble fraction of S. stercoralis L3 filariform larval antigen. The procedure was performed in accordance with the manufacturer’s recommendations.
The technicians performing the stool examinations were different from those working in serology, and all technicians knew only the results of their own tests.”
line 134 "properly" is meaningless in this context. Would "systematically" convey the intended meaning?
Response: Thank you, we have corrected this.
table 1 "depositions per day" may be better as "bowel movements per day" or "defecations per day"
Response: Thank you, we have changed depositions per day to bowel movements/day.
line 183 p values given here both appear to be for ELISA (as per table 4) and not to ELISA and stool respectively
Response: Thank you, this was a mistake. We have rewritten the sentence:
“Women living in rural areas and in houses with leaf roofing were more likely to be infected with S. stercoralis, according to ELISA results (p = 0.007 and p = 0.033, respectively).”
Table 3: please indicate in a footnote which test has been used to generate p values – I presume these are Chi square/Fisher’s exact but the statistical methods outline that Cohen’s kappa is used to compare tests
Response: Thank you for noticing this; we have included the footnote “* The statistical method used was Chi square test/Fisher’s exact test”.
